# Genome-Wide Identification of MADS-Box Family Genes in Safflower (*Carthamus tinctorius* L.) and Functional Analysis of *CtMADS24* during Flowering

**DOI:** 10.3390/ijms24021026

**Published:** 2023-01-05

**Authors:** Yifei Wang, Hengshuo Ge, Naveed Ahmad, Jia Li, Yijin Wang, Xinyi Liu, Weican Liu, Xiaowei Li, Nan Wang, Fawei Wang, Yuanyuan Dong

**Affiliations:** 1College of Life Sciences, Jilin Agricultural University, Changchun 130118, China; 2Joint Center for Single Cell Biology, School of Agriculture and Biology, Shanghai Jiao Tong University, Shanghai 200240, China

**Keywords:** *Safflower* (*Carthamus tinctorius* L.), MADS-box, phylogenetic analysis, expression patterns, flowering

## Abstract

Safflower is an important economic crop with a plethora of industrial and medicinal applications around the world. The bioactive components of safflower petals are known to have pharmacological activity that promotes blood circulation and reduces blood stasis. However, fine-tuning the genetic mechanism of flower development in safflower is still required. In this study, we report the genome-wide identification of MADS-box transcription factors in safflower and the functional characterization of a putative CtMADS24 during vegetative and reproductive growth. In total, 77 members of MADS-box-encoding genes were identified from the safflower genome. The phylogenetic analysis divided *CtMADS* genes into two types and 15 subfamilies. Similarly, bioinformatic analysis, such as of conserved protein motifs, gene structures, and cis-regulatory elements, also revealed structural conservation of *MADS*-*box* genes in safflower. Furthermore, the differential expression pattern of *CtMADS* genes by RNA-seq data indicated that type II genes might play important regulatory roles in floral development. Similarly, the qRT-PCR analysis also revealed the transcript abundance of 12 *CtMADS* genes exhibiting tissue-specific expression in different flower organs. The nucleus-localized CtMADS24 of the AP1 subfamily was validated by transient transformation in tobacco using GFP translational fusion. Moreover, CtMADS24-overexpressed transgenic Arabidopsis exhibited early flowering and an abnormal phenotype, suggesting that CtMADS24 mediated the expression of genes involved in floral organ development. Taken together, these findings provide valuable information on the regulatory role of CtMADS24 during flower development in safflower and for the selection of important genes for future molecular breeding programs.

## 1. Introduction

The MADS-box transcription factor superfamily has been widely identified in fungi, animals, and plants [1]. Members of this TF family have been shown to play fundamental roles in plant vegetative and reproductive growth, especially during flowering [2,3]. Genes encoding MADS-box proteins contain a typical domain that recognizes CArG-box, which consists of 56–58 amino acids at the N-terminus known as the MADS domain [4]. Based on their conserved MADS domains, this large gene family can be classified into two different groups, namely, type I and type II. In plants, type I members contain one (M) domain which can be subdivided into three subfamilies including, M α, M β, and M γ. On the other hand, the type II members contain three other typical domains including a less-conserved intervening (I) domain, which is crucial to the formation of DNA dimers; a well-conserved keratin (K) domain, involved in protein–protein interactions due to the coiled-coil structure; and a variable C-terminal (C) domain. On the basis of sequence divergence of the I domain, type II members can be classified into the MIKC and Mδ subgroups [5,6], whereas it can be classified into 12 groups based on their phylogenetic relationships in Arabidopsis and other model plants [7].

To date, several studies on the *MADS-box* gene family have been conducted at the genome level in many plant species; however, relatively less research has been conducted on the functions of Mδ and type I genes [8]. Mδ and type I genes have been shown to participate in the *A. thaliana* male and female gametophytes. Recent studies suggested the role of type I MADS-box genes in the regulation of endosperm development in Arabidopsis and grasses [9,10,11]. Similarly, MIKC-type members of MADS-box have been widely studied for their ability to form transcription factor complexes and their vital role in the control of flowering development [12,13,14]. According to the widely used ABCDE model of flower organ development, class A genes participate in sepal development in the first whorl; then, class A, B (AP3 and PI), and E (AGL3 and SEP) genes collectively control the petal identity in the second whorl, while class B, C (AG), and E genes all together control the stamen identity in the third whorl. Furthermore, class C and E genes combinedly determine the formation of carpel in the fourth whorl. Finally, class D (SHP1, SHP2, AGL11, and AGL13) and E genes jointly determine the formation of the ovule [15]. Several other studies demonstrated that MADS-box proteins form heterodimers with other proteins in order to regulate the plant flowering process. For example, AGL1, AG, and AP1 form homodimers, while AP3 and PI, and AP1 and CAL form heterodimers. Apart from that, different kinds of polyplexes are also involved in different stages of flowering in plants [16,17].

Over the course of plant evolution, the MADS-box family has undergone numerous variations, but their primary role of governing the flower development and morphology of flowers has remained unchanged. MADS-box genes including APETALA1 (AP1), CAULIFLOWER (CAL), and FRUITFULL (FUL) have been shown to play important roles in floral meristem specification. The AP1 (APETALA1) genes have been widely identified in flowering plants of angiosperms. Members of AP1 are mainly involved in floral organ formation and fruit and sepal development [18]. Studies on Arabidopsis have shown that the AP1 genes have undergone multiple duplications during evolution, of which the most critical was the divergence of AP1 and FUL clades [19,20,21]. Structurally, genes evolved from the same ancestor showed a high similarity and minor difference in the C-terminal amino acid arrangement. However, the C-terminus of FUL family members has an MPPWML-like hydrophobic motif, and in contrast, AP1 family members showed mutated motifs due to frameshift mutations [22,23,24]. Hence, the structural differences in these genes may lead to functional differentiation. In model plants, the AP1 gene is expressed at low levels in the floral meristem and at high levels in the developing sepals. On the other hand, *AP1* mutants cause abnormal morphology in cauliflower-like petals and turn sepals into leaf-like organs [25,26]. Similarly, *FUL* branching genes are first expressed during the transition to inflorescence meristem and later in the carpel and fruit, mainly regulating flowering transition, inflorescence configuration, cauline leaf growth, compound leaf development, fruit development, and ripening [27,28,29]. The identification and functional analysis of AP1 in different species could provide important insights to further study its molecular mechanism regulating flower development.

Safflower (*Carthamus tinctorius* L.) is an important oilseed crop widely known for its high economic and medicinal value around the world. Previous studies demonstrated the genome-wide identification of several gene and transcription factor families in safflower. Gene families such as cytochrome P450 [30,31,32,33], HCT [34], and oleosin [35] and transcription-factor-encoding families such as MYB [36], bHLH [37], and bZIP [38] have been extensively studied in safflower. However, the molecular mechanisms underlying flowering control in safflower are still unknown; only a few studies have shown that the yield of petals and seeds of safflowers is related to the number of rosettes and the number of florets in the main and lateral stem, but no information is available on the genome-wide identification and functional analysis of *MADS-box* genes in safflower. Therefore, in this study, we investigated the identification, structural and functional diversification, conserved topology, and ectopic expression profiling of the CtMADS-box gene family in safflower. Moreover, the overexpression of a candidate CtMADS24 gene was shown to promote early flowering in transgenic Arabidopsis, possibly by regulating the expression of key genes interconnected with flower development pathways. Together, these results provide a fundamental understanding of the molecular mechanism underlying flower development in safflower.

## 2. Results

### 2.1. Phylogenetic Analysis of MADS-Box Family Members

A phylogenetic tree of both type I and type II MADS-box proteins was constructed to determine the evolutionary relationships among 77 *Carthamus tinctorius* MADS-box proteins, 23 *Helianthus annuus* MADS-box proteins, and 101 *Arabidopsis thaliana* MADS-box proteins. As shown in Figure 1, the members of safflower MADS-box were divided into two types and 15 subfamilies according to the established model for plant classification. The three subfamilies of type I included M α, M β, and M γ, while the 12 subfamilies of type II included M δ, FLC, PI, AP3, AP1, AGL6, SEP, SVP, B-sister, AGL17, SOC1, and AG/STK. Approximately, 60% of safflower MADS-box family members were distributed in type II and were subdivided into 11 subfamilies, with three members in SEP (class E), four members in AG/STK (class C/D), three members in AP3/PI (class B), and five members in AP1 (class A). Compared with *Arabidopsis*, the CtMADS-box family was more closely related to sunflower, a member of the Asteraceae family, indicating that the gene family may have diverged in function during evolution. Meanwhile, structural similarities among different branches were found in AP3/PI, AP1/FUL, AGL6, and SEP subfamily members, suggesting that the members of those subfamilies may have evolved from the same ancestor with a functional relationship.

### 2.2. Analysis of the Conserved Motifs and Gene Structure

The conserved motifs of CtMADS proteins were identified with the MEME motif search program. In total, 10 conserved motifs were identified, and these motifs were further annotated with SMART protein analyzing software (Figure 2). Motifs 1 and 8 contained the typical domain of MADS. Motifs 3, 4, and 9 contained another K-box domain. Meanwhile, CtMADS01 and the other 12 type II proteins only contained the K-box domain. These structural changes may cause corresponding functional changes. In addition, most of the members showed strong conservation in the N-terminus region and relatively large differences in the C-terminus region, which is similar to that reported for MADS-box family proteins in other plants, suggesting that this family of genes is highly conserved among different plant species.

To further explore the structural differences and conservative topology of MADS-box genes, the intron and exon arrangements were determined by comparing the full-length CDS and DNA sequences of candidate CtMADS using the GSDS web server (Figure 3). The result showed no introns or very few introns in type I members, which may have caused the genetic diversity in the three subfamilies or the differences in the genetic structure of their ancestors. On the other hand, most type II genes showed more than five introns. Fewer genes showed no introns, such as *CtMADS22* and *CtMADS23*, which may have been caused by the short sequence and the loss of intron-rich K-box domains during evolution. The results showed that the genes in the same subfamily shared the same intron patterns while only having length differences. It is suggested that the subfamilies of the safflower MADS-box family have high functional similarity, while different subfamilies may have different regulatory functions.

### 2.3. Analysis of Cis-Regulatory Elements in the Promoters of MADS-Box Family Genes

To analyze the cis-regulatory elements in the promoter region of *CtMADS* genes, the 2000 bp upstream sequence of each *CtMADS* gene was extracted and analyzed using the PlantCare online web server, and the location of cis-elements is visualized in Figure 4. Nearly all the *CtMADS*-box gene promoters contained cis-acting elements related to growth and development, plant hormones, and stress responsiveness. Specifically, typical auxin response elements and gibberellin elements were found in the promoter regions of *CtMADS01*, *CtMADS24*, and *CtMADS25*. Furthermore, some protein binding sites, such as MYB and AT-rich binding sites, in the *CtMADS* promoter region were also found, suggesting that the expression of MADS-box genes may be regulated by other transcription factors. Interestingly, MeJA-related elements were observed only in type II members, indicating that some type II genes could respond to the signals of MeJA. These findings provide important insights into the structural conservation of MADS-box genes in safflower, which could be involved in crucial biological processes such as plant growth and development, responses to various stresses, and hormonal regulation.

### 2.4. Gene Expression Analysis by Transcriptome Data of CtMADS during Safflower Floral Development

To explore the potential role of the safflower MADS-box family, the expression patterns of *MADS-box* genes in different stages of safflower petal development and different tissues were investigated using RNA-seq data (Figure 5). The results showed that type I and Mδ subfamily genes were expressed at low levels in different organs and were not expressed in some organs, indicating the relatively redundant functions of those genes during the safflower developmental process. On the other hand, genes that demonstrated higher expression levels in each stage of petal development were likely to be involved in flower development. *CtMADS20*, *CtMADS24*, *CtMADS25*, and *CtMADS26*, presumed to be part of the ABCDE model, and *CtMADS14* and *CtMADS15*, which belong to the FLC subfamily, were all highly expressed in the whole process of flower development. In addition, genes such as *CtMADS39* and *CtMADS43* showed specific expression patterns in leaves, stems, and roots, indicating that these MADS-box family genes may play a regulatory role in the vegetative development of safflower.

### 2.5. Gene Expression Analysis of CtMADS in Different Flower Organs

To further elucidate the possible regulatory role of safflower MADS-box family genes in flower organ development, twelve MADS-box genes were selected based on the ABCDE model. The expression levels of these genes during flower development were investigated using qRT-PCR analysis as shown in Figure 6. The results suggested that *CtMADS24* (*class A* gene) and *CtMADS26* (*class E* gene) both had high expression patterns in sepals, and *CtMADS18* and *CtMADS20* (*class B* genes) were mainly expressed in the pistil and sepals, while *CtMADS47* (*class C* gene) was highly expressed in stamens and the pistil, suggesting that the different classes of genes have tissue-specific expression patterns during floral organ formation. Meanwhile, genes belonging to classes A, B, and E all showed high expression levels in sepals, indicating that the formation of sepals could be regulated by these three classes of genes.

### 2.6. Subcellular Location of CtMADS24

Importantly, the candidate gene CtMADS24 was detected with demonstrated ectopic expression and obvious phenotype change, and therefore we chose this gene for further functional analysis. In order to determine the subcellular localization of CtMADS24 (AP1/FUL subfamily member), the PROTCOMP online tools were initially used for computational prediction. The results showed a clear nuclear localization signal, indicating that CtMADS24 could be localized and expressed in the nucleus. Then, the experimental validation was carried out using CtMADS24-GFP fusion construct with the tobacco transient transformation system. The result of the GFP fluorescent protein was observed with a laser confocal microscope. As shown in Figure 7, the CtMADS24-GFP recombinant vector mainly appeared as a green fluorescent signal in the nucleus region when compared with the empty vector. This verified the predicted results, suggesting that the protein encoded by CtMADS24 was mainly expressed in the nucleus.

### 2.7. Ectopic Expression of CtMADS24 and Expression Level of Flowering-Related Genes in Arabidopsis

To further investigate the possible function of CtMADS24, ectopic expression analysis was carried out by introducing the cDNA of the *CtMADS24* gene driven by cauliflower mosaic virus *35S* promoter into Arabidopsis. In total, ten CtMADS24-overexpressed transgenic lines were obtained, of which two with significant phenotype changes and relatively consistent expression levels, line 1 and line 4, were selected for further analyses (Figure 8A). Compared with the wild-type lines, the flowering time of the two transgenic lines was accelerated from 35 days to approximately 27 days (Figure 8B). The rosette leaves showed a significant reduction in number, with smaller and thinner leaf morphology (Figure 8C,D). Meanwhile, the sepals of the transformed lines seemed not to completely enclose the internal floral organ (Figure 8E). Interestingly, the siliques were bumpier and smaller, with a lower seed setting rate, and the petals remained attached (Figure 8F,G). Furthermore, the differential expression of genes involved in the flower development of Arabidopsis were investigated using qRT-PCR analysis (Figure 8H). The results showed that the significantly upregulated genes such as *LFY*, *FT*, and *AGL6* may act as positive regulators during flower development in the transgenic lines. On the other hand, the *SVP* gene was downregulated and could be identified as a flowering inhibitor gene. However, genes such as *AP3* and *AP1*, which are related to floral organogenesis, remained fundamentally unchanged, while *SEP3*, which is directly related to sepal development, was significantly increased. These findings suggested that the overexpression of CtMADS24 could not cause changes in floral organs except for the sepals but induced the expression level of other endogenous flowering genes in Arabidopsis to regulate flower development.

## 3. Discussion

Flower development containing reproductive growth, flowering time regulation, and vegetative growth monitoring are only a few of the many roles that have been ascribed to MADS-box genes in flowering plants. As a result of these findings, evolutionary scientists have focused on MADS-box genes and their role in flower development and regulation. In recent years, with the accomplishment of various whole genome sequencing projects, the study of related gene families is rapidly progressing. Many MADS-box genes have been identified in various plant species including Arabidopsis [21], rice [39], sunflower [40], bamboo [41], pear [42], wheat [43], strawberry [44], and many other plants. These findings laid the groundwork for studying the evolution of MADS-box genes. However, their regulatory roles were not consistent and were found to be unique to particular species. Therefore, the identification and analysis of MADS-box genes in different kinds of species are necessary to gain deeper insights into the function of MADS-box genes.

In this study, a total of 77 MADS-box genes were identified from the safflower genome after excluding the missing and incomplete sequences. The numbers of MADS-box genes varied between different species: safflower (77) and sunflower (23) had less than Arabidopsis (101) and rice (75). This indicated that Asteraceae plants might have lost MADS-box genes during the evolutionary process. The phylogenetic analysis showed that MADS-box family members could be divided into two main types and 15 subfamilies according to the established model for plant classification [21]. In terms of gene structure and protein conserved motifs, the conservation within the same family was significant, but there were large structural differences among different families as found in marigold [45] and bamboo [41]. In the process of higher plant evolution, the conservation and number of introns in the plant type I MADS-box family were relatively low, while type II MADS-box family members with the K-box structure were more conservative and richer in introns, similar to wheat and orchid MADS-box genes [5,6,46,47]. Furthermore, the distribution of the cis-elements in the promoter region of CtMADS genes showed the presence of important elements such as responses to plant growth and developmental processes and hormones or environmental stimuli. Consistently with previous results, safflower type II genes had highly conserved motifs both interspecific and intraspecific.

The discovery of genes related to flower development and flowering transition remains a major focus in plant research. Studies have shown that MADS-box genes belonging to the type I clade are mainly involved in the development of the female gametophyte, the embryo, and seeds, while plant type II MADS-box genes have specific roles in flower organ development, especially in specifying floral organ identity [39,48]. In most plants, different subfamily members often have different expression patterns [45,49]. In our study, the transcript abundance in the flower development of type I clade genes was extremely low, while the subfamilies belonging to the type II clade displayed diverse expression patterns, and most of the type II genes were expressed in the reproductive organs, thereby indicating that type II MADS genes might play important roles in flower development in safflower. To better understand the role of the type II clade in the flower development of safflower, we performed a comprehensive investigation with the expression assay of these genes in different flower organs. The ABCDE model is a classic model of plant flower development [50,51]. Our findings indicated that several classes of MADS-box genes such as A, B, C, and E were expressed at varying levels during flower development, signifying their individual roles in floral organ development in safflower. On the other hand, genes with identical expression patterns may work together in floral organ development. These findings were found to be consistent with previous studies on plant species including carnation [52] and orchid [53].

Floral development from vegetative to reproductive development is an important transition process in plants, which involves the interaction of many genes. The AP1 subfamily is an ancient evolutionary lineage, and members of this family are mainly involved in floral organ formation, fruit development, and sepal development [54,55,56,57]. To date, AP1 orthologs have also been isolated in a wide range of plant species, such as *Gossypium hirsutum* [58], *Magnolia wufengensis* [59], and *Petunia* [60]. As already described in class A genes of the flowering model, the identification and analysis of genes in this class are significant for the in-depth understanding of the flower development process. In safflower, one of the homologous genes of the AP1 subfamily (CtMADS24) was identified and further investigated for functional analysis. The protein encoded by CtMADS24 was localized in the nucleus of *Nicotiana benthamiana* leaves. Moreover, the expression pattern of CtMADS24 in transgenic plants was notably higher during the flowering development period in sepals than in other tissues, which is in line with its expected function in sepal identity specification. Our results also showed that the ectopic expression of CtMADS24 in Arabidopsis induced early flowering by promoting the expression of other signaling genes involved in floral transition; however, no changes in the expression were found in petal characteristic genes. Previous studies on Arabidopsis, *Fagopyrum esculentum*, *Rosa chinensis*, and other plant species also suggested that AP1 overexpression results in significantly earlier flowering [61,62,63]. In addition, early reproductive development often leads to insufficient nutrition in plants, which in turn leads to a decrease in seed yield and even abortion [64]. Transgenic Arabidopsis showed significantly smaller siliques, suggesting that the ectopic expression of CtMADS24 accelerates the flowering process, inhibits vegetative development, and may lead to plant abortion as discussed by [45]. Together, our findings provide novel insights into the regulatory mechanisms of floral organ development by the intervention of MADS24 transcription factor, thereby facilitating future molecular breeding research on safflower.

## 4. Materials and Methods

### 4.1. Plant Materials

The safflower variety of “Ji Hong No. 1” was grown in the experimental field of Jilin Agricultural University in Changchun, Jilin Province, China. Petal tissues were collected at the bud, early flower, full, and decline stages of flowering. The leaves, sepals, stamens, and pistils were collected for tissue-specific expression analysis. All plant materials were snap-frozen in liquid nitrogen and stored at −80 °C until next use.

### 4.2. Identification of MADS-Box Family Genes

The safflower genome sequence was deposited at NCBI under the bioproject accession number: PRJNA399628. The MADS-box transcription factor family sequences were obtained from the safflower genome. The amino acid sequences of Safflower MADS-box family members were shown in Appendix A. The HMM file along with the MADS domain (PF00319) from the Pfam protein database (http://pfam.xfam.org, Pfam 31.0, accessed on 10 June 2021) was downloaded and then searched for the MADS-box candidate genes in the safflower genome with the HMMER3.0 server using e-values lower than 0.01 and the default parameters. The MADS-box core sequences were confirmed using the SMART database and the NCBI CDD web server (http://www.ncbi.nlm.nih.gov/Structure/cdd/wrpsb.cgi, accessed on 10 June 2021). Finally, the redundant sequences were removed to ensure that each sequence contained at least one structurally representative region of the MADS-box. The MADS-box sequence of Arabidopsis and sunflower were obtained from the Plant Transcription Factor Database (http://planttfdb.gao-lab.org/, accessed on 10 June 2021).

### 4.3. Phylogenetic Analysis, Conserved Motif, and Gene Structure Analysis of CtMADSs

The downloaded sequences were further screened using the methods of Ayaz et al. [65], and multiple sequence alignment of MADS-box protein sequences from Arabidopsis, sunflower, and safflower was analyzed with Muscle software. A phylogenetic tree was constructed using the maximum likelihood (ML) method with 1000 iterations for the bootstrap values. Tree files were viewed and edited using ITOLS online tools (https://itol.embl.de/, accessed on 20 June 2021). Similarly, the conserved protein motifs were analyzed by using the full-length protein sequences of candidate CtMADS through MEME software (http://meme-suite.org/tools/meme, accessed on 22 June 2021) with the following parameters: the maximum value of motifs was set to identify 10 motifs, the minimum motif width was 6, and the maximum motif width was 200. The identified conserved protein motifs were further annotated by SMART database and the NCBI CDD web server (http://www.ncbi.nlm.nih.gov/Structure/cdd/wrpsb.cgi, accessed on 22 June 2021). Afterward, the gene and coding DNA sequences (CDS) of safflower were obtained from safflower genome data, and the intron and exon distribution of the candidate *CtMADSs* were analyzed and compared using GSDS web server (http://gsds.cbi.pku.edu.cn/, accessed on 30 June 2021). The above analyses were carried out according to the method of Ayaz et al. [66].

### 4.4. Prediction of Cis-Acting Elements in Promoter Sequences

A 2000 bp upstream sequence from the translation initiation codon of each *CtMADS* gene was obtained from the genomic data of safflower in order to explore the distribution of cis-acting elements in the promoter region of *CtMADS* genes. The online web server of PlantCARE (http://bioinformatics.psb.ugent.be/webtools/plantcare/html/, accessed on 30 June 2021) was exploited to locate and identify various cis-elements in *CtMADS*-*box* genes and then visualized using TBtools software [67].

### 4.5. Safflower MADS-Box Genes Expression Pattern Analysis

The fragments per kilobase of transcript per million fragments (FPKM) values were obtained from the safflower RNA-seq data (PRJNA909037) to analyze the probable transcript abundance of CtMADS genes in different floral organs of safflower (bud stage, early flowering stage, full flowering stage, decline stage) and different vegetative tissues (root, leaf, stem). All the obtained values were log2-transformed and visualized by TBtools software.

### 4.6. RNA Isolation and Quantitative Real-Time PCR Analysis

The expression levels of 12 genes in three flower tissues and leaf tissue were analyzed by qRT-PCR analysis. For this purpose, the total RNA content was extracted using the Fast-Pure Plant Total RNA Isolation Kit (Vazyme, Nanjing, China), and the extracted RNA was measured for purity and concentration by a NanoDrop 2000 ultraviolet spectrophotometer. Then, the first-strand cDNA was synthesized using MonScript™ RTIII All-in-One Mix with dsDNase (Monad, Suzhou, China) kit and preserved at −20 °C. A set of gene-specific primers of safflower CtMADS genes was synthesized based on the information obtained from the coding sequences in Premier version 6 (Premier Biosoft, Palo Alto, CA, USA). Primer sequences were shown in Appendix A. To verify the integrity of primer specificity, each primer pair was generated away from the conserved domain of the genes. The internal reference genes of safflower used in the qRT-PCR analysis included the *18S* ribosomal *RNA* gene [37]. All qRT-PCR reactions were performed with MonAmp™ ChemoHS qPCR Mix from Monad Biotechnology Co., Ltd. (Suzhou, China) using the Stratagene MX3000P real-time PCR machine. Following the manufacturer’s instructions, a 20 μL reaction mixture was prepared with 2.0 μL cDNA template, 0.2 μM primers (F/R), 0.2 μL ROX dye (100×), 10 μL master mix, and 7 μL RNase-free water. PCR conditions were set according to the manufacturer’s protocol. The relative transcript abundance values were calculated using the 2^−ΔΔCt^ method [68].

### 4.7. ORF Cloning and Subcellular Localization

The open reading frame of the *CtMADS24* (without a stop codon) was amplified from cDNA and cloned into pCAMBIA1302 using the MonClone™ Hi-Fusion Cloning Mix V2 kit. Primer sequences were shown in Appendix A. Subcellular localization was performed by transient expression of the CtMADS24-GFP proteins in *Nicotiana benthamiana* leaves according to previously established methods [69]. The recombinant plasmids pCAMBIA1302-CtMADS24-GFP and pCAMBIA1302-GFP were transformed into *Agrobacterium tumefaciens* GV3101(pSoup-19) competent cells by the freeze–thaw method. Transformed GV3101(pSoup-19) cells were injected into the leaves of *Nicotiana benthamiana* for transient expression. After being cultured for 60 h under long-day conditions of 16 h light, the cells were observed with a laser confocal microscope (Leica TCS SP8, Wetzlar, Germany).

### 4.8. Generation of CtMADS24-Overexpressed Transgenic Arabidopsis Lines and Ectopic Expression Analysis

The wild-type Arabidopsis plants were grown in the artificial growth chamber of our lab under long-day (LD) conditions (16 h light/8 h dark) at 21 °C. For the construction of overexpression vectors, the ORF of *CtMADS24* was cloned into the pCAMBIA3301 vector and transformed into *Agrobacterium tumefaciens EHA105* competent cells.The transformation was carried out by the floral dip method [70] The gene-specific primers were used to screen out transgenic plants until T3 generation was obtained harboring the CtMADS24 gene. The phenotype of the transgenic plants was observed for morphological differences, and the expression profiling was carried out simultaneously using qRT-PCR assay. All the primers sequences were shown in Appendix A.

## 5. Conclusions

In this study, a comprehensive and systematic analysis of the safflower MADS-box transcription factor family was conducted for the first time. In total, 77 members of the MADS-box transcription factor family were extensively identified in the safflower genome. The phylogenetic relationships, gene structures, conserved motifs, and cis-elements were analyzed using bioinformatic analysis. The expression profile of the MADS-box family genes in different developmental stages and flower organs revealed that they are likely involved in the floral organ development of safflower. In addition, the overexpression of a candidate gene, CtMADS24, caused early flowering and abnormal sepal development in Arabidopsis. These findings provide comprehensive resources for the improvement of new safflower cultivars with early flowering and increased petal production in future molecular breeding programs.

## Figures and Tables

**Figure 1 ijms-24-01026-f001:**
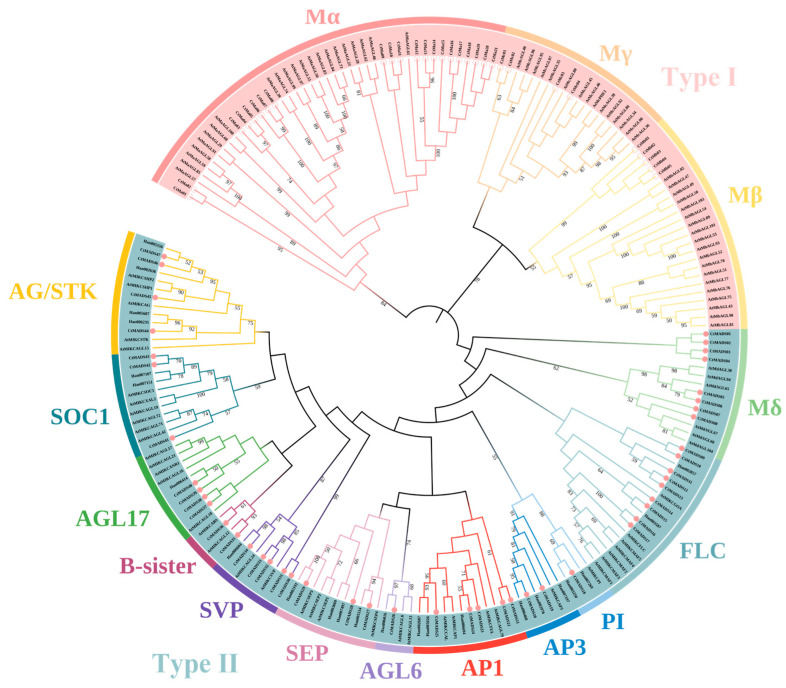
Phylogenetic analysis of safflower, sunflower, and Arabidopsis MADS-box family proteins. A total of 77 safflower MADS-box proteins, 23 Sunflower MADS-box proteins, and 101 Arabidopsis MADS-box proteins were included in the phylogenetic tree constructed using a maximum likelihood method. The red circles represent the MADS-box proteins in safflower.

**Figure 2 ijms-24-01026-f002:**
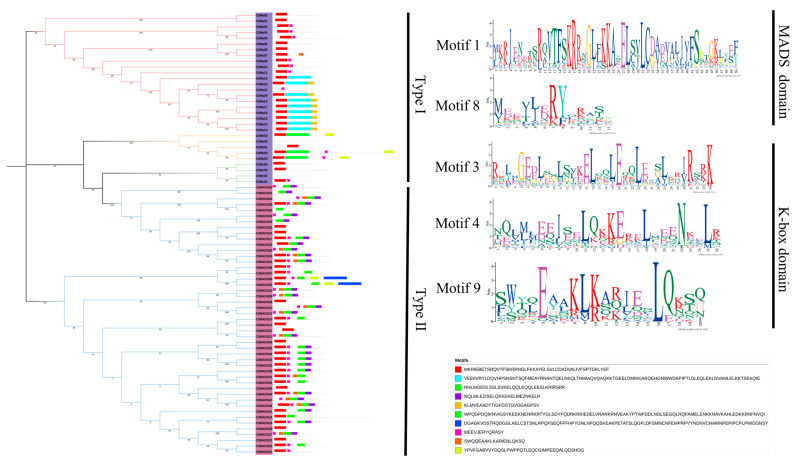
Distribution and sequences of signature motifs of MADS and K-box structural domains obtained from safflower MADS-box proteins using MEME analysis. Different motifs are represented by different colors. Typical MADS and K-box structural domains extracted from MADS-box protein sequence are also shown on the right side.

**Figure 3 ijms-24-01026-f003:**
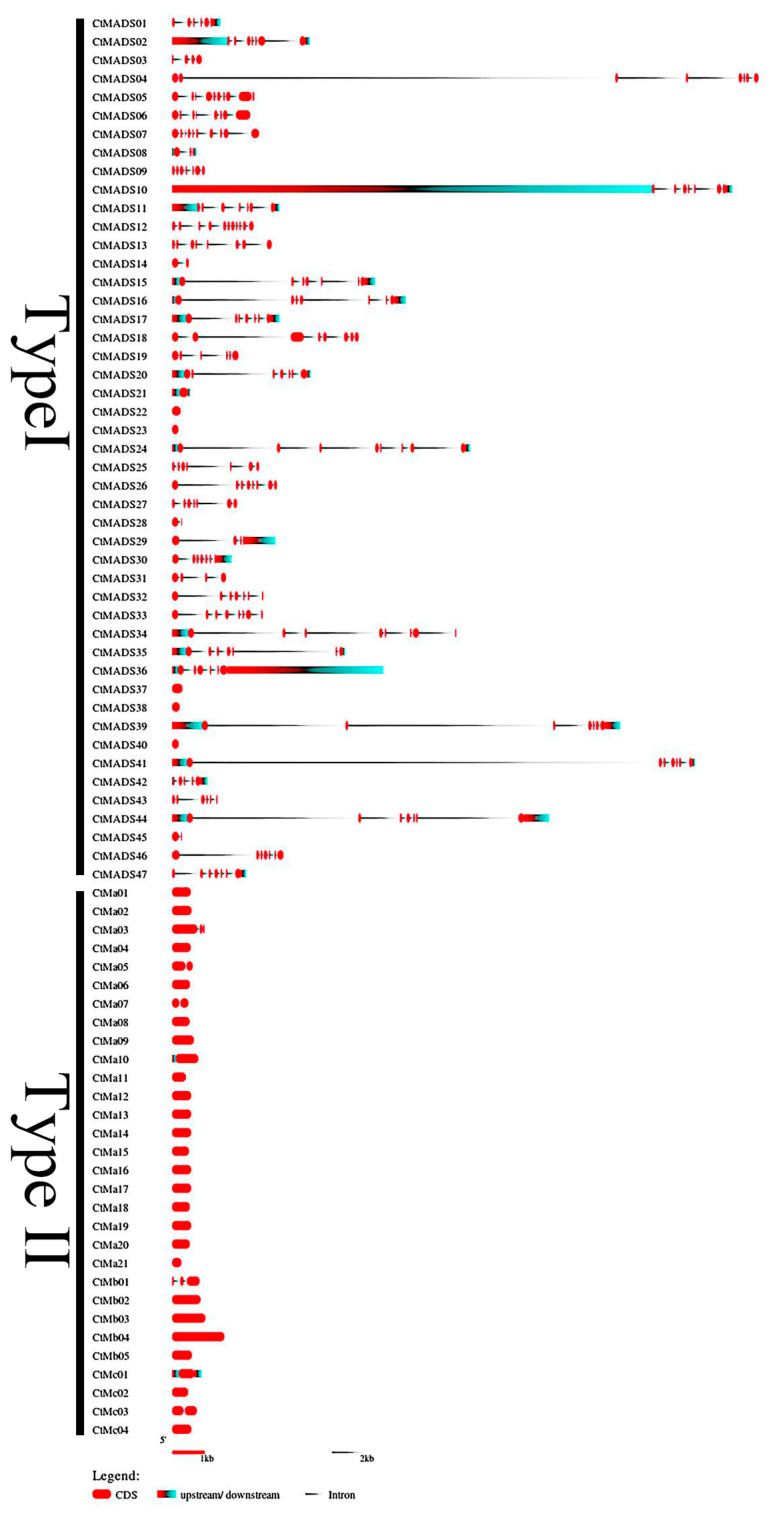
The organization of exon–intron structures of *CtMADS* genes. The lines indicate introns, the red rectangles indicate exons, and the blue rectangle indicate UTRs.

**Figure 4 ijms-24-01026-f004:**
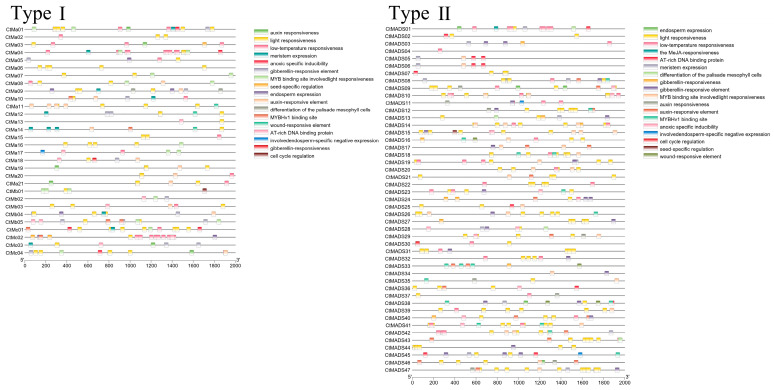
Distribution of cis-acting regulatory elements of *CtMADS* genes in safflower. The different colored boxes represent the presence of specific cis-regulatory elements in the promoters of *CtMADS-box* genes.

**Figure 5 ijms-24-01026-f005:**
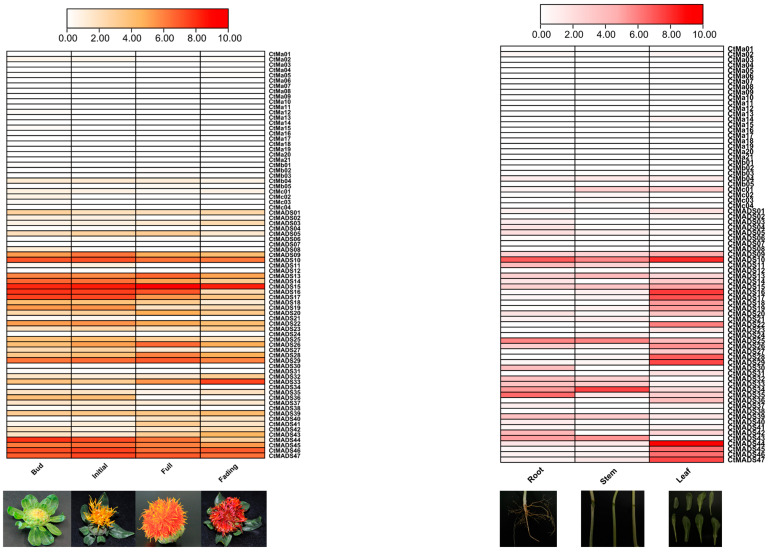
Expression analysis of MADS-box genes in safflower. The FPKM values of the safflower MADS-box genes were log2-transformed to create the heatmap using TBtools software. The white and red colors represent the expression levels of *CtMADS* genes from low to high.

**Figure 6 ijms-24-01026-f006:**
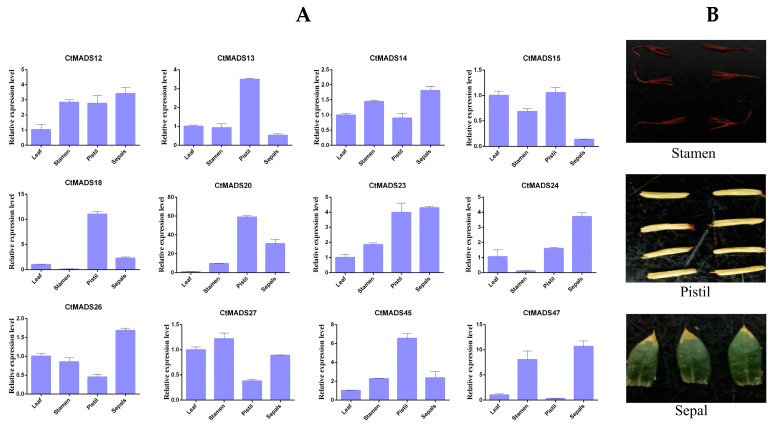
Organ-specific expression analysis of 12 *CtMADS* genes in safflower. (**A**) The genes were classified into different classes including class A, B, C, and E and four different organs (leaf, stamen, pistil, and sepal) were selected for expression analysis. Error bars represent the standard deviation *n* = 3. (**B**) The phenotype of different safflower flower organs used in the study.

**Figure 7 ijms-24-01026-f007:**
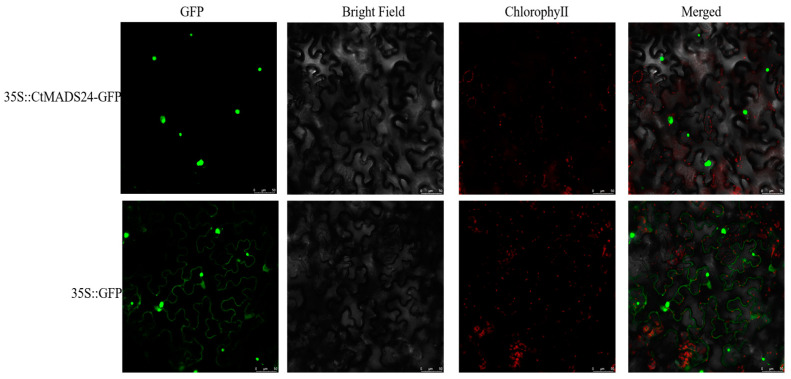
Subcellular localization of CtMADS24 using transient expression system in tobacco leaves. The CtMADS24-GFP (green fluorescent protein) fusion construct was localized to the nucleus. The fluorescence signal was detected with a laser scanning confocal microscope. GFP indicates fluorescence of green fluorescent protein, and the red color shows the auto-fluorescence of chlorophyll. Scale bar = 50 μm.

**Figure 8 ijms-24-01026-f008:**
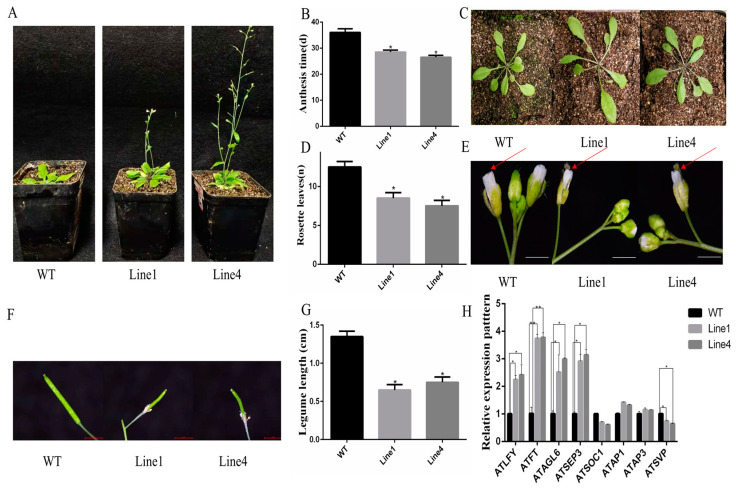
Phenotypic variations and expression of key floral regulator genes in *CtMADS24*-overexpressed transgenic Arabidopsis. (**A**) The flowering phenotype of *CtMADS24* transgenic Arabidopsis and WT plants. (**B**) Statistics for anthesis time of *CtMADS24* transgenic Arabidopsis and WT plants. (**C**) The rosette phenotype of *CtMADS24* transgenic Arabidopsis and WT plants. (**D**) Statistics for rosette leaves of *CtMADS24* transgenic Arabidopsis and WT plants. (**E**) The sepal phenotype of *CtMADS24* transgenic Arabidopsis and WT plants, bar = 1 cm. The red arrows pointed the internal floral organ in WT and transgenic Arabidopsis lines. (**F**) Legume morphology of *CtMADS24* transgenic Arabidopsis and WT plants, bar = 0.5 cm. (**G**) Statistics for legume length of *CtMADS24* transgenic Arabidopsis and WT plants. (**H**) The expression pattern of key genes involved in flowering development (* significant difference at *p* < 0.05, ** significant difference at *p* < 0.01).

## Data Availability

Not applicable.

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
