# Peer review of "Genome-Wide Identification of MADS-Box Family Genes in Safflower (Carthamus tinctorius L.) and Functional Analysis of CtMADS24 during Flowering"

_ijms, 2023, doi:10.3390/ijms24021026_

Round 1
Reviewer 1 Report
In this manuscript, authors identified the MADS-box transcription factors family in the Safflower, and CtMADS24 was found that ectopic overexpression in Arabidopsis resulted in early flowering and an abnormal phenotype. I found this topic interesting but I have many concerns related to the manuscript. In my opinion, introduction is well-written. However, part of the methods section requires detailed information. There are still good rooms to improve the quality of the manuscript. I recommend the authors to have all my comments addressed and revise the manuscript.
1. Line 13-18, this sentence is too long, I propose to amend it.
2. Line 39-41, I think a reference should be added to this sentence.
3. Line 58-79, the author should introduce the relationship between AP1 subfamily and MADS-box family, the unacquainted reader will be confused.
4. Line 80-81, this sentence needs to be corrected. the scarce genome information?
5. Figure2, analysis of conserved amino acid motifs: There is a large variation in the number of conserved motifs (ranged from 1 to 5) for CtMADS family. Is it the same in other species?
6. Fingure 3, some putative genes possess very long introns (ie: CtMADS04, CtMADS39 and CtMADS41). Is it the same for the corresponding orthologous genes in other species (Arabidposis, Sunflower)? Is the exon-intron structure for the CtMADS gene family overall similar with the species cited in the manuscript?
7. Line 161-163, 167-169, make sure you present only results in the Result Section. You must not explain why you got those results or provide other research data to support your data. This needs to be in the Discussion section. Please check the entire manuscript for similar situations.
8. Figure 5, expression profile of CtMADS genes across in different stages of safflower petal development and different tissues: Some of putative CtMADS genes are not expressed (ie: CtMa04, CtMa05 etc….). It is not clear if there is a basal expression or no corresponding mRNAs were detected in the transcriptome dataset. Are they pseudo-genes? Finally, is CtMADS family consists of 77 genes?
9. I also encourage the authors to explore chromosomal location of the CtMADS family, and micro and macro synteny of the CtMADS gene clusters to a few species of plants since these genes arise from tandem or segmental duplications.
10. The resolution of Figure 2, 3 and 4 is not high enough, it is blurry.
11. The legend of Figure 6 need to be more accurate, especially the three pictures on the right.
12. In 2.6, can the authors explain why CtMADS24 was used for follow-up analysis, in my opinion, CtMADS14 and CtMADS26, etc, were superior to CtMADS24.
13. Line 262-274, I suggest that authors discuss information, such as the number, gene structure and protein conserved motifs of MADS gene families in safflower and other similar species, which is more like a summary description of the results.
14. Line 308-317, is this work only reported in Arabidopsis and whether it is found in other species? I suggest that the authors add more information to confirm this result.
15. In 4.2, the safflower genome, how was it obtained? A reference should be added to the genome.
16. In 4.6, all RNA sequencing data from this study need to deposited in the NCBI or other public platform under certain accession number.
17. In 4.7, can the authors explain why 18s was used as an internal reference gene? Please provide statistics for validation of 18S as reference gene.
Author Response
In this manuscript, authors identified the MADS-box transcription factors family in the Safflower, and CtMADS24 was found that ectopic overexpression in Arabidopsis resulted in early flowering and an abnormal phenotype. I found this topic interesting but I have many concerns related to the manuscript. In my opinion, introduction is well-written. However, part of the methods section requires detailed information. There are still good rooms to improve the quality of the manuscript. I recommend the authors to have all my comments addressed and revise the manuscript.
Response: The authors would like to thank the reviewer for providing his valuable suggestions. All of your comments are very important and of great significance to improve the scientific content of our manuscript.
Point 1: Line 13-18, this sentence is too long, I propose to amend it.
Response 1: According to your valuable suggestions, we have rephrased these sentences in the Abstract section during revisions. Please refer to (page 1; lines 14-26).
Point 2: Line 39-41, I think a reference should be added to this sentence.
Response 2: Thank you for your suggestion. The reference has been added to the “Introduction” of revised version (page 2; lines 51-52).
Point 3: Line 58-79, the author should introduce the relationship between AP1 subfamily and MADS-box family, the unacquainted reader will be confused.
Response 3: We feel very sorry for the mistake. Following your suggestion, the relationship between the AP1 (APETALA1) and MADS-box superfamily has been added to the “Introduction” of the revised version (page 2; lines 75-77).”
Point 4: Line 80-81, this sentence needs to be corrected. the scarce genome information?
Response 4: We feel very sorry for this lapse. The sentence has been corrected in the “Introduction” of the revised version (page 3; lines 101-105).”
Point 5: Figure2, analysis of conserved amino acid motifs: There is a large variation in the number of conserved motifs (ranged from 1 to 5) for CtMADS family. Is it the same in other species?
Response 5: We thank the reviewer for this rigorous point, the large variation in the number of conserved motifs were also found in other species such as Sunflower, Carnation and Arabidopsis. Hence, the variation of conserved motifs of MADS-box could also be conserved in safflower. Please refer to the following bibliography. Chrysanthemum (Doi:10.1007/s13205-019-1897-z).
(Doi: 10.1016/j.compbiolchem.2020.107424),
Carnation (Doi: 10.3390/genes9040193)
Rice (Doi:10.1186/1471-2164-8-242)
Arabidopsis (Doi:10.1105/tpc.011544).
Point 6: Figure3, some putative genes possess very long introns (ie: CtMADS04, CtMADS39 and CtMADS41). Is it the same for the corresponding orthologous genes in other species (Arabidopsis, Sunflower)? Is the exon-intron structure for the CtMADS gene family overall similar with the species cited in the manuscript?
Response 6: Thank you for commenting on this. The intron length of the corresponding orthologous genes in Arabidopsis (Doi:10.1105/tpc.011544), Chrysanthemum (Doi: 10.1016/j.compbiolchem.2020.107424) were similar to that of safflower. Similarly, the exon-intron structure of the CtMADS gene family were consistent with MADS box in Carnation (Doi: 10.3390/genes9040193), Sesame (Doi: 10.1016/j.gene.2015.05.018) and Rice (Doi:10.1186/1471-2164-8-242). Therefore, the gene structure of MADS-box could also be conserved in safflower.
Point 7: Line 161-163, 167-169, make sure you present only results in the Result Section. You must not explain why you got those results or provide other research data to support your data. This needs to be in the Discussion section. Please check the entire manuscript for similar situations.
Response 7: We feel very sorry for the mistake; We have corrected this issue following your comments. The discussion and interpretation of the previous results were removed from the result section during the revision.
Point 8: Figure 5, expression profile of CtMADS genes across in different stages of safflower petal development and different tissues: Some of putative CtMADS genes are not expressed (ie: CtMa04, CtMa05 etc….). It is not clear if there is a basal expression or no corresponding mRNAs were detected in the transcriptome dataset. Are they pseudo-genes? Finally, is CtMADS family consists of 77 genes?
Response 8: Thank you for your suggestions. We agree with your point of view about the low expression and no expression of some of putative CtMADS genes in different stages of safflower petal development and different tissues. However, the transcriptome data suggested that these genes were slightly expressed in different seed development stages. Therefore, these genes are not pseudo-genes. Yes, we identified a total of 77 MADS-box genes in safflower.
Point 9: I also encourage the authors to explore chromosomal location of the CtMADS family, and micro and macro synteny of the CtMADS gene clusters to a few species of plants since these genes arise from tandem or segmental duplications.
Response 9: Thanks for your valuable suggestion. It is indeed an important aspect of this study; however, the genome sequencing of safflower lacks Hi-C sequencing and chromosome assembly to date, and therefore, it is not possible to obtain chromosomal location and synteny and collinearity analysis.
Point 10: The resolution of Figure 2, 3 and 4 is not high enough, it is blurry.
Response 10: Thank you for pointing out this lapse. The resolution of Figures 2, 3 and 4 were increased in the revised version (pages 5-6 lines 166-178 and page 7 lines 198-203).
Point 11: The legend of Figure 6 need to be more accurate, especially the three pictures on the right.
Response 11: The legend with clearer description have been added in the “Figure 6” of the revised version (pages 9 lines 244-247).
Point 12: In 2.6, can the authors explain why CtMADS24 was used for follow-up analysis, in my opinion, CtMADS14 and CtMADS26, etc, were superior to CtMADS24.
Response 12: Thanks for reviewer’s questions, the genes like CtMADS14, CtMADS26, you just mentioned, were also listed as candidate genes, but ectopic expression of CtMADS24 in Arabidopsis showed a more obvious phenotype change, and therefore, we followed up with this gene to further investigate its regulatory mechanism in flowering. According to your comment, the reason has been explained in section 2.6 during the revisions.
Point 13: Line 262-274, I suggest that authors discuss information, such as the number, gene structure and protein conserved motifs of MADS gene families in safflower and other similar species, which is more like a summary description of the results.
Response 13: Thank you for your constructive suggestion. We have incorporated these informations in the “Discussion” section of the revised version (pages 11-12 lines 321-325).
Point 14: Line 308-317, is this work only reported in Arabidopsis and whether it is found in other species? I suggest that the authors add more information to confirm this result.
Response 14: We thank the reviewer for his valuable suggestions. Following your comment, we have added the information of other related species in the “Discussion” section of the revised version (pages 12 lines 363-365).
Point 15: In 4.2, the safflower genome, how was it obtained? A reference should be added to the genome.
Response 15: Thank you. The accession number “PRJNA399628 “were cited in this and all relevant sections, which was used for obtaining the genome sequence of safflower. Please see section 4.2 in the revised version.
Point 16: In 4.6, all RNA sequencing data from this study need to deposited in the NCBI or other public platform under certain accession number.
Response 16: The RNA sequencing data in the study has been deposited in the NCBI public platform with accession number “PRJNA909037“ and was added in “Materials and methods” of the revised version.
Point 17: In 4.7, can the authors explain why 18s was used as an internal reference gene? Please provide statistics for validation of 18S as reference gene.
Response 17: Thank you for your question. We use 18S as an internal reference gene due to its invariant expression across tissues, organs and experimental treatments. The references using 18s as a reference has been added in the “Materials and methods” of the revised version (pages 14 line 452).

Reviewer 2 Report
This article presented Genome-wide identification of MADS-box family genes in Safflower (Carthamus tinctorius L.) and functional analysis of CtMADS24 during flowering. The study is well organized and data is well arranged. The findings would be helpful for future studies. Before recommending this article for publication, there are some shortcomings for that should be resolve.
Provide affiliations of all authors also number them in sequence.
Methods are not well presented in the abstract. Which techniques were used for the analysis.
Also add quantitative and specific results in this section.
Provide economic, medicinal and industrial significance of Safflower.
Also provide details of other gene families studies conducted on Safflower or MADS-box genes related genes or transcription factors studies.
Italicize plant names.
Line 85-96 the authors presented their findings. Here the authors should provide the aim of the study not the results.
Line 312-315 should be cited with relevant studies.
Section 4,3 and 4.4 should be cited with relevant studies
https://doi.org/10.3390/ijms22179175, https://doi.org/10.1016/j.plaphy.2021.01.042,
Section 4.7 should be cited with recent studies having this protocol.
https://doi.org/10.1007/s10725-021-00785-7,
Conclusion is well presented but some future recommendations should also be presented.
Author Response
This article presented Genome-wide identification of MADS-box family genes in Safflower (Carthamus tinctorius L.) and functional analysis of CtMADS24 during flowering. The study is well organized and data is well arranged. The findings would be helpful for future studies. Before recommending this article for publication, there are some shortcomings for that should be resolve.
Response: The authors would like to thank the reviewer for providing his valuable suggestions. All of your comments are very important and of great significance to improve the scientific content of our manuscript.
Point 1: Provide affiliations of all authors also number them in sequence.
Response 1: We apologize for this lapse. The affiliation of all authors has been provided and each author was numbered accordingly in the revised manuscript.
Point 2: Methods are not well presented in the abstract. Which techniques were used for the analysis.
Response 2: Thank you for pointing out this mistake. We have added the brief description of the methods and techniques used for the analysis of CtMADS-box genes within the “Abstract” section of the revised manuscript (page 1 lines 11-29).
Point 3: Also add quantitative and specific results in this section.
Response 3: Following your suggestion, we have added the core findings of our study in the “Abstract” section of the revised manuscript (page 1 lines 17-27).
Point 4: Provide economic, medicinal and industrial significance of Safflower.
Response 4: We are very thankful to the reviewer for his constructive reviews on our manuscript. According to your suggestion, the economic, medicinal and industrial significance of Safflower have been added in the background of the “Abstract” section of the revised manuscript (page 1 lines 11-13).
Point 5: Also provide details of other gene families studies conducted on Safflower or MADS-box genes related genes or transcription factors studies.
Response 5: Thank you for highlighting this point. We have added the details of other gene and transcription factor families in safflower. (page 3 lines 100-105).
Point 6: Italicize plant names.
Response 6: The scientific names of the plant species were italicized in the revised version.
Point 7: Line 85-96 the authors presented their findings. Here the authors should provide the aim of the study not the results.
Response 7: Thank you for highlighting this lapse. We have modified this part according to your suggestion please see (pages 3; lines 100-105).
Point 8: Line 312-315 should be cited with relevant studies.
Response 8: Thank you for the reminder. The relevant literature has been cited in these sentences within the revised manuscript (page 12; lines 381-383).
Point 9: Section 4,3 and 4.4 should be cited with relevant studies
Response 9: Thank you for pointing this out. Section 4.3 and 4.4 were synthesized and cited with relevant literature in the revised version. (page 13; lines 409-410 and lines 423-424).
Point 10: Section 4.7 should be cited with recent studies having this protocol.
Response 10: The relevant literature has been added in the revised manuscript (page 14; lines 458-459).
Point 11: Conclusion is well presented but some future recommendations should also be presented.
Response 11: According to your suggestion, future recommendations have been added to the “Conclusion” of revised manuscript (page 15; lines 497-499).

Round 2
Reviewer 1 Report
Based on my suggestion in the first review, the authors have addressed my concerns appropriately. I still have a few questions for the manuscript.
1. Line 263-265, I think this sentence should be put in the first sentence, so as to make it clear to the reader.
2. “The MADS-box transcription factor family sequences were obtained from searched in the safflower genome (PRJNA399628)”. The author should indicate which database owns the information?
3. The authors explain the genome sequencing of safflower lacks Hi-C sequencing and chromosome assembly to date, and therefore, it is not possible to obtain chromosomal location and synteny and collinearity analysis.
As far as I know, the genome at chromosome level of Safflower has been published in 2021, please check at doi: 10.1111/pbi.13586. In addition, if the author's previous work had not been based on this genome file (Wu, Zhihua, et al.2021), it is doubtful whether there would have been errors in the identification of gene families due to the low level of the genomes referenced. Please consider this issue carefully.
Round 3
Reviewer 1 Report
Based on my suggestion in the review, the authors have addressed my concerns appropriately. The authors did a commendable job in putting together all the safflflower available MADS-box genes resources and performing in-silico analysis, this work is meaningful. So, the manuscript can be accepted for publication.